# Animal Age Affects the Gut Microbiota and Immune System in Captive Koalas (*Phascolarctos cinereus*)

Jiayan Chen,[a] Weijie Lv,[a] Xueli Zhang,[b] Tianyou Zhang,[b] Jian Dong,[b] Zhihui Wang,[b] Tingting Liu,[b] Peng Zhang,[b] Michael Pyne,[c] Guixin Dong,[b] Shining Guo[a,d]

[a]College of Veterinary Medicine, South China Agricultural University, Guangzhou, China
[b]Chimelong Safari Park, Chimelong Group Co., Panyu, Guangzhou, China
[c]Currumbin Wildlife Hospital, Currumbin, Queensland, Australia
[d]Guangdong Research Center for Veterinary Traditional Chinese Medicine and Natural Medicine Engineering Technology, Guangzhou, China

**ABSTRACT** Gut microbiota is one of the major elements in the control of host health. However, the composition of gut microbiota in koalas has rarely been investigated. Here, we performed 16S rRNA gene sequencing to determine the individual and environmental determinants of gut microbiota diversity and function in 35 fecal samples collected from captive koalas. Meanwhile, blood immune-related cytokine levels were examined by quantitative reverse transcription-PCR to initially explore the relationship between the gut microbiota and the immune system in koalas. The relative abundance of many bacteria, such as *Lonepinella koalarum*, varies at different ages in koalas and decreases with age. Conversely, *Ruminococcus flavefaciens* increases with age. Moreover, bacterial pathways involved in lipid metabolism, the biosynthesis of other secondary metabolites, and infectious disease show a significant correlation with age. Age affects the relationship between the microbiota and the host immune system. Among them, the gut microbiota of subadult and aged koalas was closely correlated with CD8$\beta$ and CD4, whereas adult koalas were correlated with CLEC4E. We also found that sex, reproductive status, and living environment have little impact on the koala gut microbiota and immune system. These results shed suggest age is a key factor affecting gut microbiota and immunity in captive koalas and thus provide new insight into its role in host development and the host immune system.

**IMPORTANCE** Although we have a preliminary understanding of the gut microbiota of koalas, we lack insight into which factors potentially impact captive koalas. This study creates the largest koala gut microbiota data set in China to date and describes several factors that may affect gut microbiota and the immune system in captive koalas, highlighting that age may be a key factor affecting captive koalas. Moreover, this study is the first to characterize the correlation between gut microbiota and cytokines in koalas. Better treatment strategies for infectious disorders may be possible if we can better understand the interactions between the immune system and the microbiota.

**KEYWORDS** koalas, age, gut microbiota, immune system, cytokines, animal models, immune markers

Koalas (*Phascolarctos cinereus*) are solitary Australian mammals that eat virtually solely *Eucalyptus* leaves. However, *Eucalyptus* foliage is toxic or fatal to most other mammals. Koalas can break down plant matter through fermentation and enzymatic breakdown, extracting enough nutrients to keep their metabolisms running (1). On the one hand, several lignin- and tannin-degrading bacteria, such as *Streptococcus bovis* and *Lonepinella koalarum*, have been identified in the gastrointestinal tracts of koalas (2, 3). Gut microbiota is thus assumed to play a key part in this process. Koalas' ability

Address correspondence to Shining Guo, shining@scau.edu.cn, or Guixin Dong, dgx@chimelong.com.

The authors declare no conflict of interest.

to detoxify *Eucalyptus* foliage, on the other hand, may be attributable to expansions within the cytochrome P450 gene family (4, 5).

The billions of microorganisms that live in the gut, known as the gut microbiota, are now recognized as one of the major elements in the control of host health (6–8), and they contribute significantly to the development of the immune system, behavior, and a number of other elements of host biology (9–13). According to earlier studies, normal gut microbiota variation is influenced by factors, such as diet, age, sex, genetics, and environmental exposure (14–21). Because herbivorous mammals lack the genetic coding for fiber degradation enzymes, they must form a symbiotic relationship with cellulolytic microbes in their gut to assist them to meet their nutritional needs (22). Complex anaerobic microbial communities can very efficiently digest ingested plant biomass due to the evolution of specific organs in herbivorous mammals, such as the rumen, cecum, and colon (23). Gut microbiota can also help the host detoxify secondary compounds from plants (24, 25). Giant and red pandas can adapt to bamboo containing cyanide-containing compounds, and desert woodrats can adapt to the highly toxic creosote bush containing creosote.

Infectious diseases such as chlamydial disease and koala retrovirus (KoRV) infections pose a significant danger to wild koala populations (26, 27). The immunological response of koalas to infectious illness is poorly understood (28). The immune system has been found to be a protective factor during infectious diseases, and cytokines are suitable targets for assessing local and systemic immune responses to intracellular infections (29). Tumor necrosis factor alpha (TNF-$\alpha$), interleukin-6 (IL-6), and other important cytokines in the koala that can serve as indicators for Th1 and Th2 immune responses, as well as a number of cell surface receptors and markers, such as CD4, CD$\beta$, and CLEC4E, have been described in preliminary research (26, 28). Thus, we may deduce the koala immune response from blood cytokine levels. Furthermore, because the gut contains 70 to 80% of immune cells, there is a complicated interplay between the gut microbiota and the body's cells and processes (such as the immune system) (30–32).

A recent study on koalas from Australia's Featherdale Wildlife Park discovered that their gut microbiomes shift in response to a higher proportion of leaves in their meals, eventually reaching adult composition by independence (33). However, their gut microbiota shift in response to aging has rarely been investigated. Most functional investigations of the gut microbiota and immune system have thus far been limited to model organisms such as humans and mice, or laboratory animals in a controlled environment (34, 35). Few functional investigations have been conducted in nonmodel creatures, particularly endangered wild mammals. The captive koalas' fecal microbiomes were similar to those found in wild koalas (36). As a result, captive koalas could be useful for functional research. Furthermore, research on gut microbiota and blood cytokine levels provides more specific insights into the koala's unique biology without harming or disturbing an endangered animal. Study of the koala gut microbiota could provide better feeding regimens that take into account the species' reproductive characteristics, as well as nutritional supplements such as probiotics.

We analyzed the gut microbiota composition in 35 fecal samples and the cytokine levels in 29 blood samples from the koalas living in the Guangzhou Chimelong Safari Park in China. We hypothesized that the gut microbiota of captive koalas is mainly influenced by age variation and describe the composition of the gut microbiota in different age groups. We also tried to investigate the possible interactions between the host immune system and the microbiota, as well as how age may affect these interactions. In addition, we analyzed whether sex, reproductive state, and living environment affected captive koalas.

## RESULTS

**The koala gut microbiota.** We discovered 12,741 amplicon sequence variants (ASVs) using deep 16S rRNA gene amplicon sequencing in 35 fecal samples (after rarefaction, mean $\pm$ the standard deviations [SD] ASVs/sample = 768.54 $\pm$ 132.94; range, 466

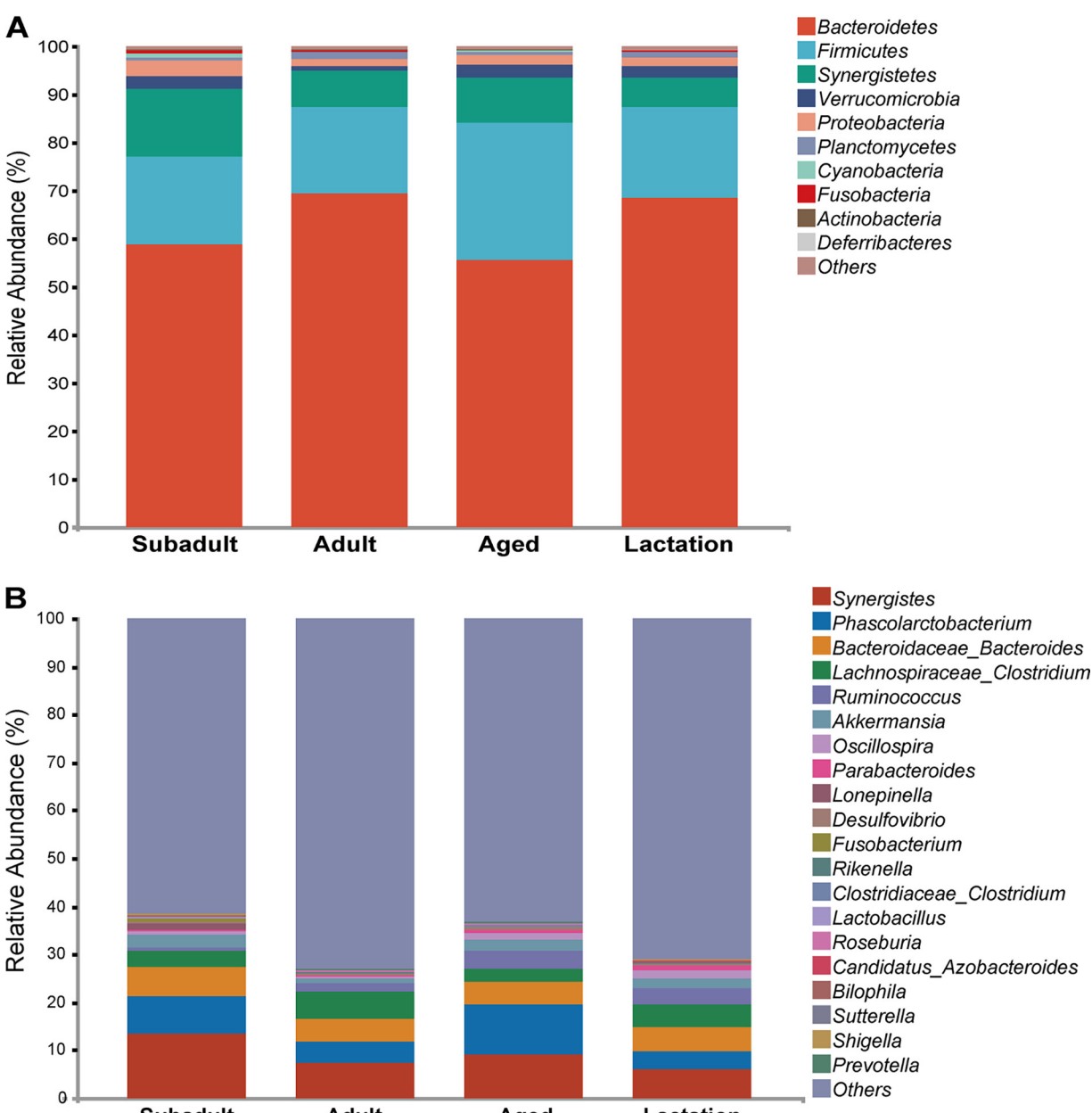

FIG 1 Taxonomic composition of the koala gut microbiota at phylum and genus levels. (A) Ten most abundant phyla. (B) Twenty most abundant genera.

to 1,046). There were 12,741 ASVs from 31 phyla, 85 classes, 151 orders, 255 families, 486 genera, and 639 species among them. *Bacteroidetes* are the most numerous phyla at the phylum level (Fig. 1A), followed by *Firmicutes* and *Synergistetes*, in each sample. These three phyla were responsible for 93% of all sequences. The "core microbiota" of the koala gut microbial community, which made up 31.69% of the total ASVs, consisted of 12 genera (2.47% of the total identified genera) that were each present in 100% of the samples: *Synergistes* (abundance, 8.99%), *Phascolarctobacterium* (6.62%), *Bacteroidaceae_Bacteroides* (5.14%), *Lachnospiraceae_Clostridium* (4.09%), *Ruminococcus* (2.33%), *Akkermansia* (2.16%), *Oscillospira* (0.93%), *Parabacteroides* (0.6%), *Desulfovibrio* (0.32%), Fusobacterium (0.29%), Rikenella (0.11%) and Lactobacillus (0.11%) (Fig. 1B).

**Age variation in gut microbiota diversity and composition.** To better describe the dynamic gut microbiota alterations of the captive koala, we investigated the relationship of ASVs (average relative abundance across samples of ≥0.01%) with noticeably

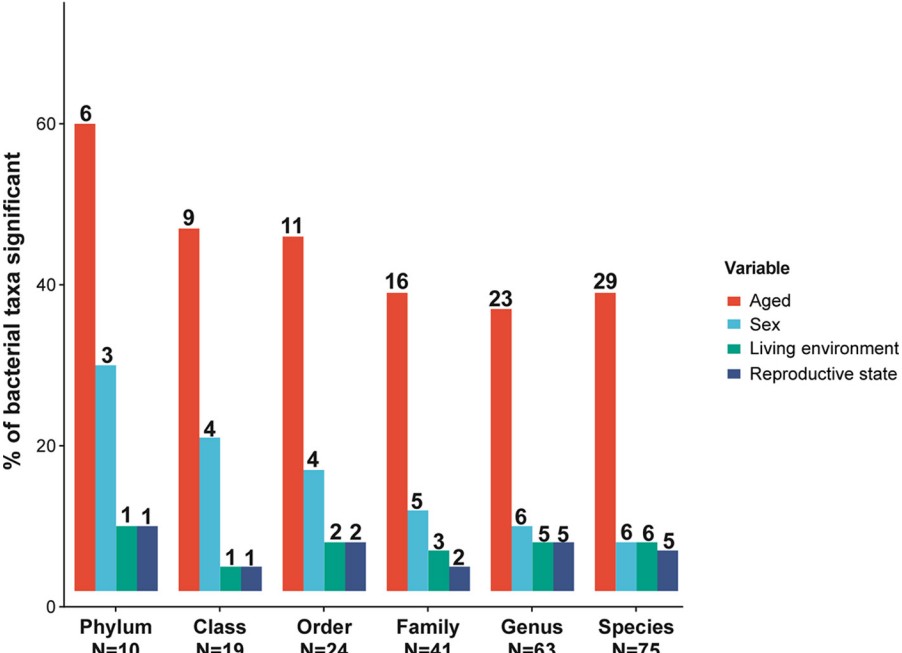

**FIG 2** Age exerts the strongest effect on bacterial relative abundance. The percentages of taxa that are significantly associated with age, sex, living environment, or reproductive state, across six taxonomic levels are shown. The significance of the difference was verified by using a Kruskal-Wallis rank sum test (or Mann-Whitney U test [for two-sample groups]). Only taxa with $P$ values <0.05 were considered significant. The numbers below the bars indicate the total taxa measured per level, while the numbers above depict the number of taxa significantly differentially abundant.

varied abundance among age groups with age as a continuous variable. To begin, we found 6 phyla, 9 classes, 11 orders, 16 families, 23 genera, and 29 species that were significantly differentially abundant in different age groups (Fig. 2). Thus, age significantly correlated with 37% of the bacterial genera examined (and 39 to 60% of taxa at other taxonomic levels [Fig. 2]) and predicted the relative abundance of gut microbiota at all taxonomic levels.

The Spearman correlation was then employed to determine the relationship between their abundance and age. *Bacteroidetes*, *Synergistetes*, *Verrucomicrobia*, and *Fusobacteria* were negatively linked with age at the phylum level (Fig. 3A and Table 1), whereas *Firmicutes* and *Actinobacteria* were positively associated with age. Thirteen genera were adversely linked with age at the genus level (Fig. 3B and Table 1), the majority of which were possible commensals. Three *Bacteroidales* genera (identified as *S24-7*, unclassified *Bacteroidales*, and unidentified *Bacteroidales*), two *Lactobacillales* genera (*Lactobacillus* and unclassified *Lactobacillaceae*), two *Fusobacteriales* genera (*Fusobacterium* and unidentified *Fusobacteriaceae*), unidentified *YS2*, *Sutterella*, *Lonepinella*, *Synergistes*, and *Akkermansia*. Five genera from the order *Clostridiales* (unidentified *Lachnospiraceae*, *Clostridium*, *Oscillospira*, *Phascolarctobacterium*, and unclassified *Clostridiales*), two genera from the order *Burkholderiales* (*Delftia* and *Oxalobacter*), unidentified *Coriobacteriaceae*, *Rikenella*, and unidentified *Streptophyta* were also positively associated with age. Furthermore, compared to subadults and adults, the aged had a significantly decreased *Bacteroidetes*/*Firmicutes* (B/F) ratio (Fig. 3C). The B/F ratio was highest in adults (median = 4.48); in subadults it was slightly lower than in adults (median = 3.57), and it decreased in the aged (median = 1.94).

Linear discriminant analysis effect size (LEfSe) was used to find differential gut bacterial taxa that demonstrated the highest abundance in each of the three age groups (Fig. 3D). The diversity of gut microbiota differed significantly in each age group, which was consistent with the microbial composition data. At the phylum level, *Fusobacteria*, *Synergistetes*, and *Verrucomicrobia* showed the highest abundance in subadults, *Bacteroidetes* in adults, whereas *Actinobacteria* and *Firmicutes* showed the highest

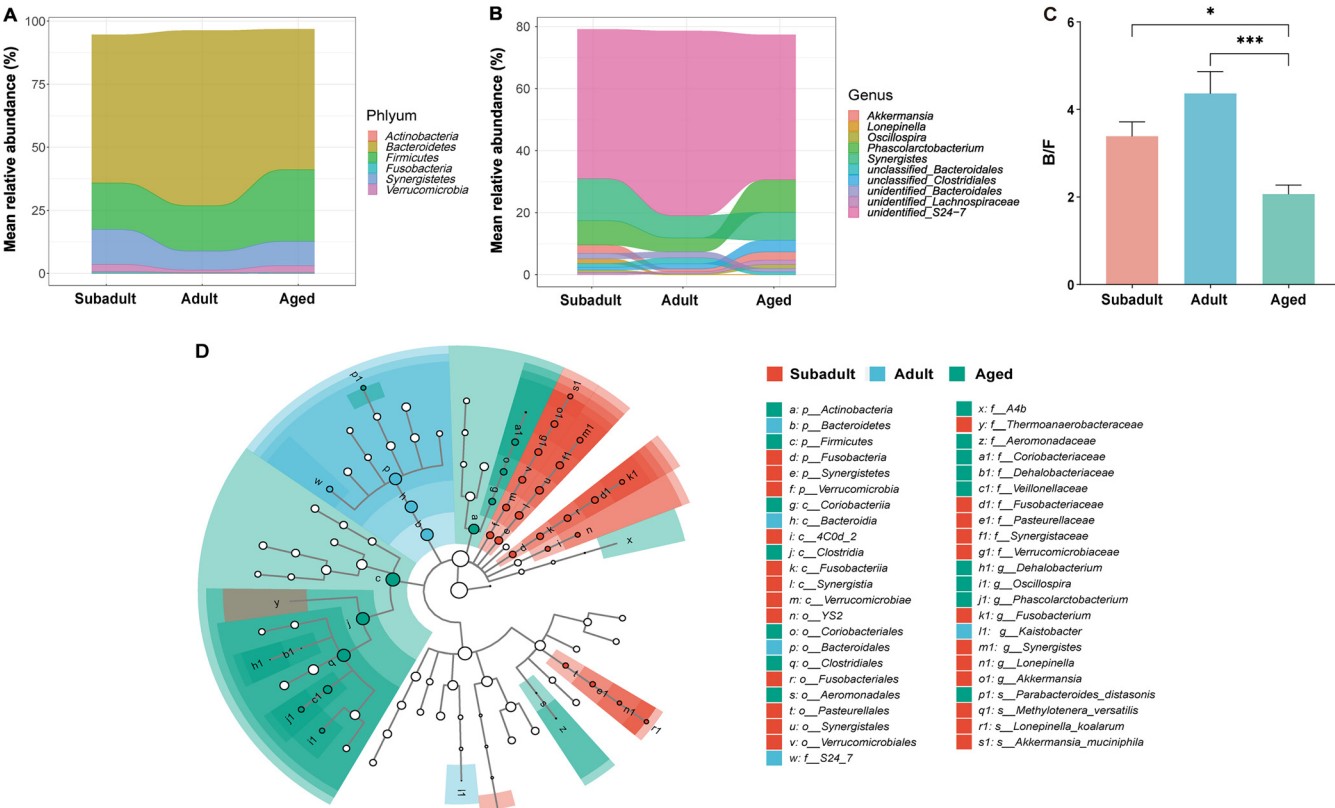

**FIG 3** Correlation between differential abundant gut microbiota and age. (A) Six differential abundant taxa at the phylum level. (B) Top 10 differential abundant taxa in the genus. (C) Relative proportions of the *Bacteroidetes/Firmicutes* (B/F) ratio. (D) LEfSe analysis generated differences in the abundance of the bacterial taxa of three age groups ($P < 0.05$, LDA > 2). Mean values $\pm$ the standard errors of the mean (SEM) are shown. The significance of the difference between groups was tested by the nonparametric Kruskal-Wallis test. *, $P < 0.05$; ***, $P < 0.001$.

abundance in the aged. The gut microbiota of subadults and the aged had the most families and genera among the three age groups. The dominating genera in subadults were *Fusobacterium*, *Synergistes*, *Lonepinella*, and *Akkermansia*. Three genera showed the highest abundance in the aged: *Dehalobacterium*, *Oscillospira*, and *Phascolarctobacterium*. Only the genus *Kaistobacter* was highly enriched in the adults.

After that, we focused on the age-characterized bacterial taxa (species). In this study, *Lonepinella koalarum* was discovered and found in high relative abundance (average, >0.3%).

**TABLE 1** Significant correlation modeling between differentially abundant and age

| Species | Spearman *r* | *P* |
|---|---|---|
| p_Actinobacteria | 0.522 | 0.004 |
| p_Firmicutes | 0.526 | 0.003 |
| p_Fusobacteria | −0.715 | <0.001 |
| g_unidentified_Coriobacteriaceae | 0.527 | 0.003 |
| g_unidentified_Bacteroidales | −0.503 | 0.005 |
| g_unidentified_YS2 | −0.423 | 0.022 |
| g_Lactobacillus | −0.647 | <0.001 |
| g_unidentified_Lachnospiraceae | 0.454 | 0.013 |
| g_unclassified_Clostridiales | 0.486 | 0.007 |
| g_Allobaculum | −0.456 | 0.013 |
| g_Fusobacterium | −0.774 | <0.001 |
| g_unidentified_Fusobacteriaceae | −0.625 | <0.001 |
| g_Sutterella | −0.517 | 0.004 |
| g_Delftia | 0.425 | 0.021 |
| g_Oxalobacter | 0.413 | 0.026 |
| g_Lonepinella | −0.605 | 0.001 |

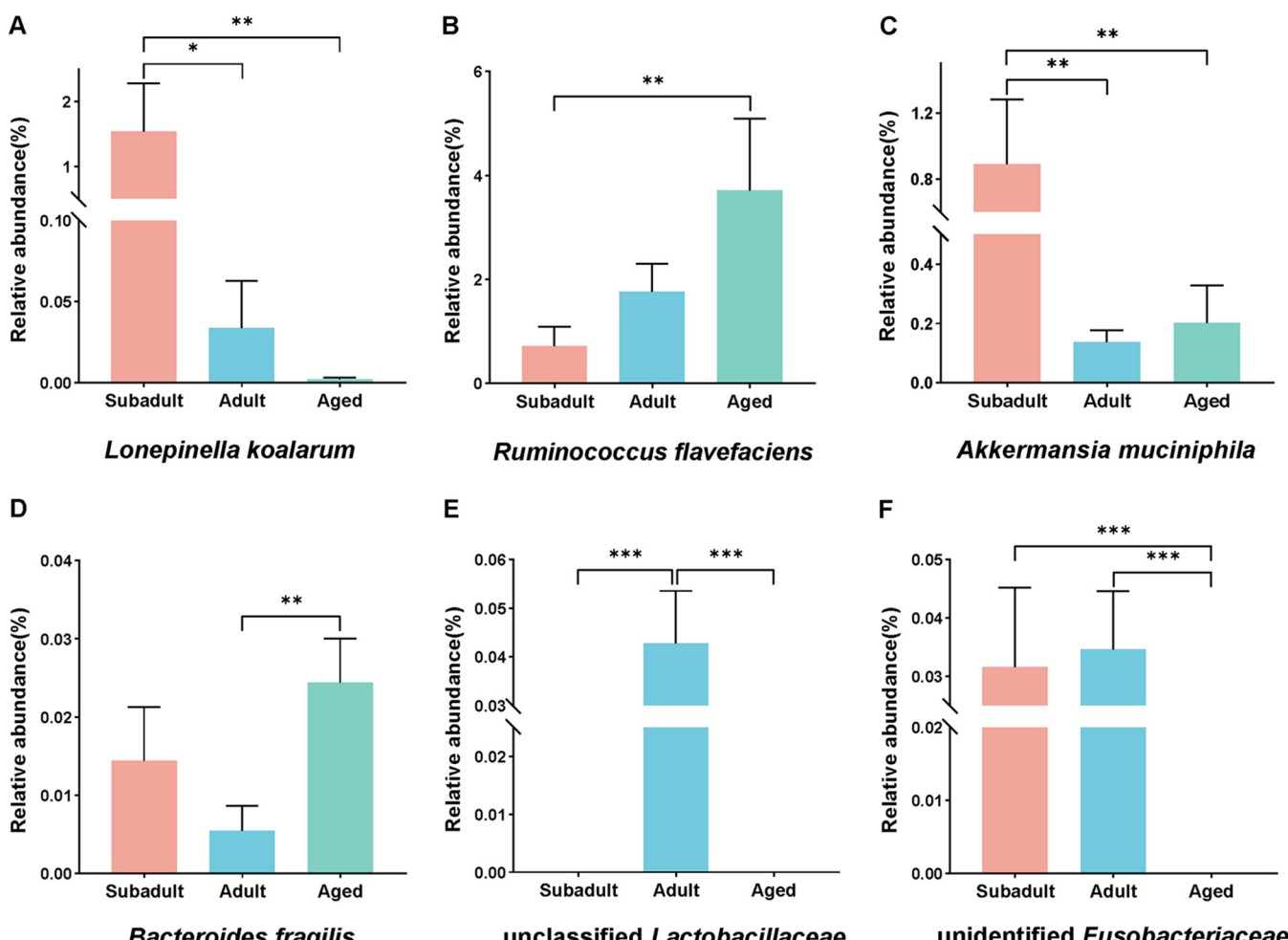

**FIG 4** Relative abundances of bacterial taxa in six species that are significantly associated with age. (A) *Lonepinella koalarum*. (B) *Ruminococcus flavefaciens*. (C) *Akkermansia muciniphila*. (D) *Bacteroides fragilis*. (E) Unidentified *Fusobacteriaceae*. (F) Unclassified *Lactobacillaceae*. Mean values ± the SEM are shown. The significance of the difference between groups tested by using a nonparametric Kruskal-Wallis test with a Bonferroni *post hoc* test. *, $P < 0.05$; **, $P < 0.01$; ***, $P < 0.001$.

Subadult koalas had the most *L. koalarum*, and there was a negative connection with age (Fig. 4A). *Ruminococcus flavefaciens*, on the other hand, was positively related to age, with the largest abundance in aged koalas (Fig. 4B). The abundance of *Akkermansia muciniphila* was higher in subadults, while *Bacteroides fragilis* was higher in aged koalas (Fig. 4C and D). We then discovered two unclassified or unidentified bacteria that were not detected in each subadult and aged koala (Fig. 4E and F). Unclassified *Lactobacillaceae* was only found in the feces of adults, whereas unidentified *Fusobacteriaceae* was found in the feces of subadults and adults but not the aged.

Chao1 and observed species demonstrated significant differences in bacterial richness among the three age groups at the level of alpha diversity (Fig. 5A). Subadults and adults had significantly higher bacterial richness than aged koalas, but there was no significant difference between subadults and adults. Subadults and aged clustered differentially in the principal coordinate analysis (PCoA) based on the Bray-Curtis distance matrix for beta diversity (Fig. 5B). Furthermore, the results of a permutational multivariate analysis of variance (PERMANOVA) (Table 2) based on the Bray-Curtis distance revealed significant variations across the three age groups ($P = 0.001$).

The functional profile of the gut microbiota is predicted by age. We identified 33 Kyoto Encyclopedia of Genes and Genomes (KEGG) modules (see Fig. S2) to be significantly associated with age using the software Phylogenetic Investigation of Communities by Reconstruction of Unobserved States (PICRUSt). The bacterial pathways at level 1 of KEGG

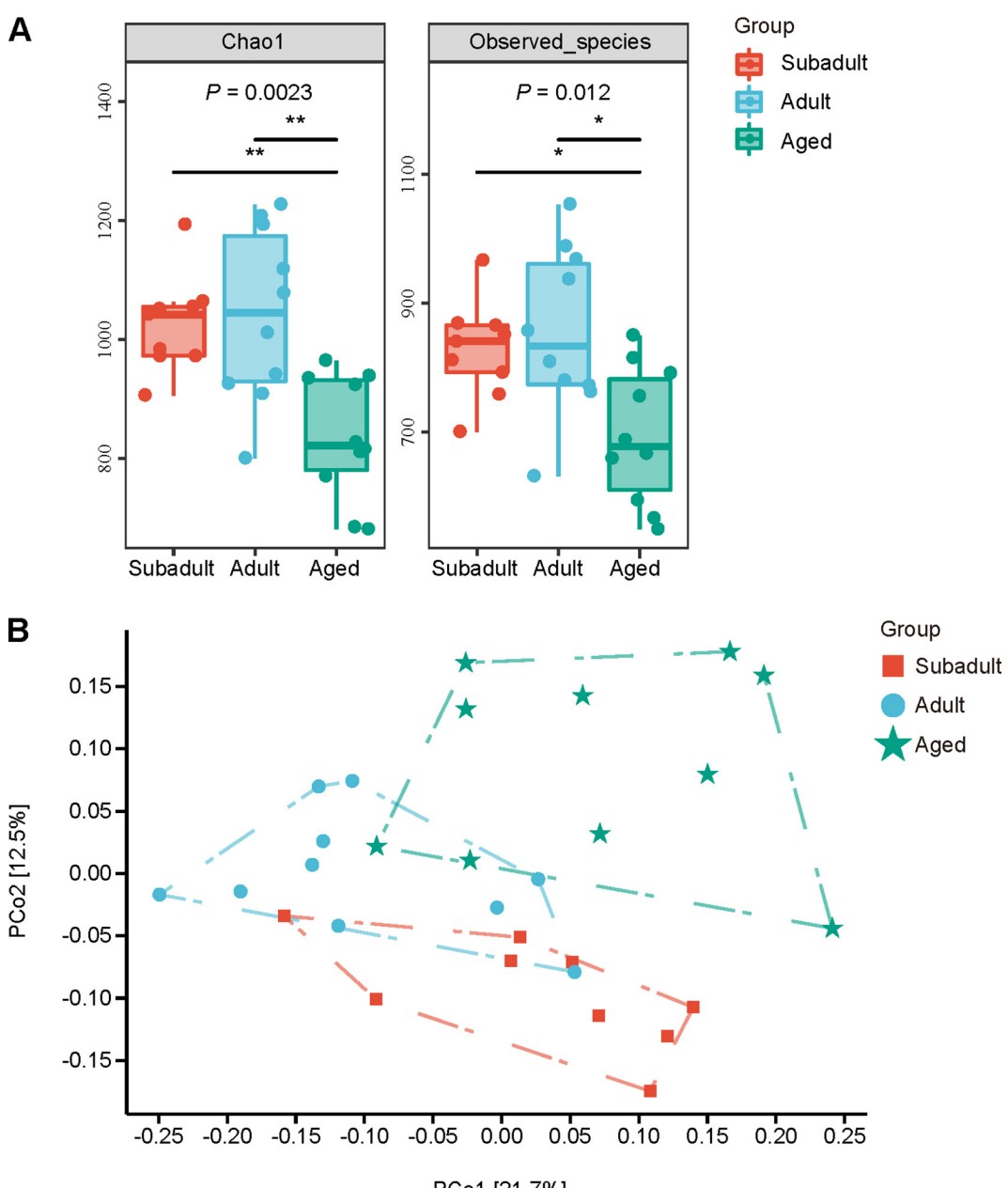

**FIG 5** Diversity differences in the gut microbiota among the three age groups. (A) Alpha-diversity index (Chao1 index and observed species diversity). (B) Principal coordinate analysis (PCoA). The significance of the difference between groups tested by using a nonparametric Kruskal-Wallis test with a Bonferroni *post hoc* test. *, $P < 0.05$; **, $P < 0.01$.

Orthology (KO) were mainly enriched in the metabolism pathway, followed by genetic information processing and cellular processes. Bacterial functional pathways were mainly enriched in amino acids, carbohydrates, cofactors, and vitamins in metabolism at KO level 2.

Compared to the subadult koala, lipid metabolism (e.g., primary and secondary bile acid biosynthesis) showed higher proportions in the adult koala (Fig. 6A), while infectious diseases (e.g., epithelial cell signaling in *Helicobacter pylori* infection) showed higher proportions in the aged koala (Fig. 6B). Twelve pathways differed between adult and aged koalas (Fig. 6C). Of these, the largest significant differences were pathways for amino acid metabolism, lipid metabolism, and replication and repair. Amino acid metabolism (e.g., the metabolism of cysteine, methionine, arginine, proline, and histidine, and the biosynthesis of valine, leucine, isoleucine, lysine, phenylalanine, tyrosine,

**TABLE 2** PERMANOVA of host factors and environment in the koalas' data set[a]

| Factor | $R^2$ (%) | $P$ |
|---|---|---|
| Age | 20.34 | 0.001 |
| Sex | 5.12 | 0.11 |
| Reproductive state | 9.11 | 0.568 |
| Living environment | 3.62 | 0.418 |

[a]$n = 35$ samples.

and tryptophan) and replication and repair (e.g., homologous recombination, DNA replication, base excision, and repair of nucleotide excision and mismatch) showed higher proportions in the aged koala, while lipid metabolism (e.g., fatty acid degradation, sphingolipid metabolism, and primary and secondary bile acid biosynthesis) showed higher proportions in the adult koala. Furthermore, three microbial functions, including lipid metabolism, the manufacture of various secondary metabolites, and the pathways for infectious diseases, were found to significantly correlate with age. The proportions of lipid metabolism and biosynthesis of other secondary metabolites pathways in the adult group were significantly higher than in the subadult and aged group, but there

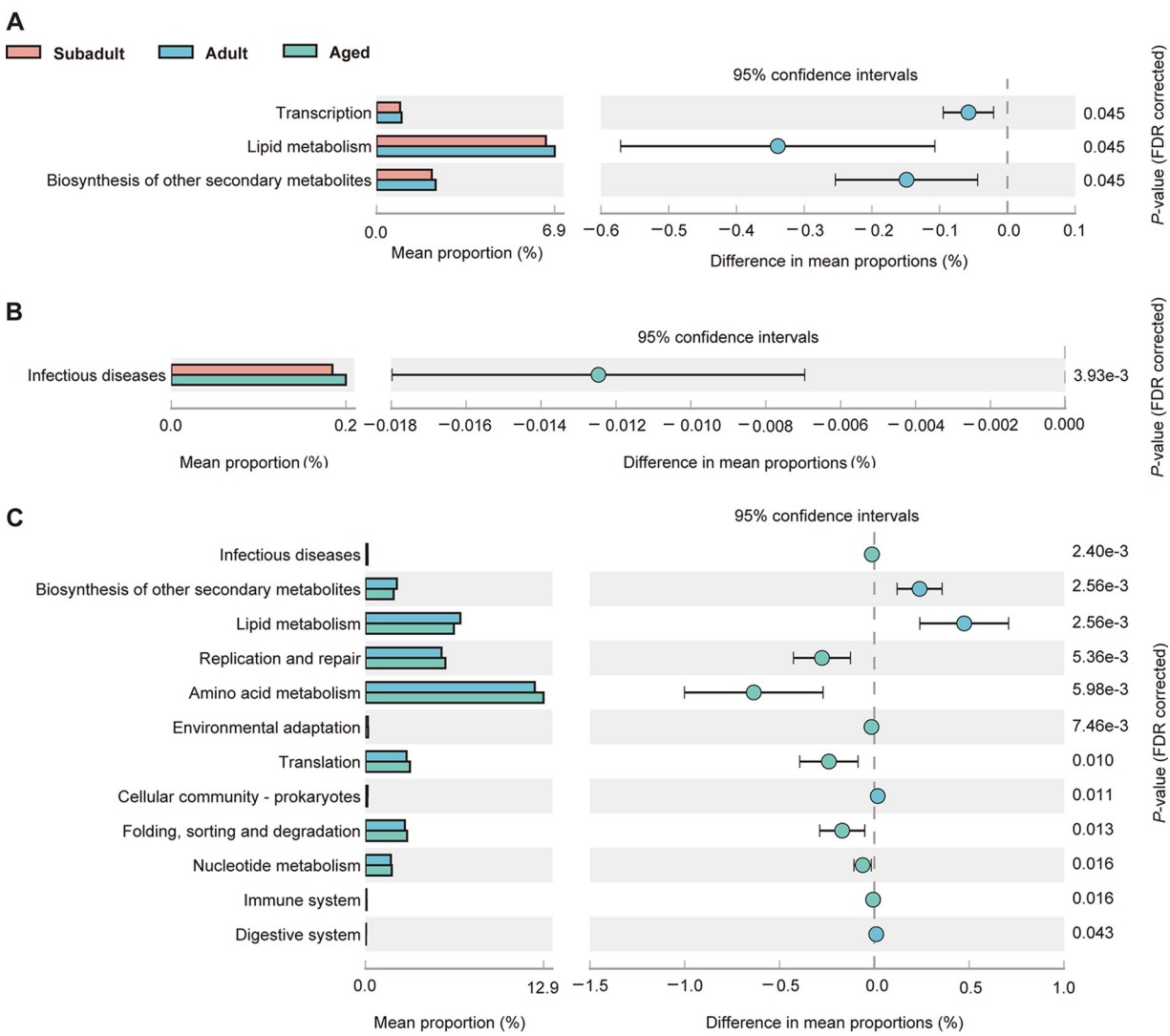

**FIG 6** Age predicts the functional profile of the gut microbiota. (A to C) Bacterial pathways at level 2 of KEGG Orthology (KO) differ in proportions in three age groups: subadult compared to adult (A), subadult compared to aged (B), and adult compared to aged (C). The bar plot shows the mean proportions of differential KEGG pathways predicted using PICRUSt2. The difference in proportions between the groups is shown with 95% confidence intervals. Only $P$ values <0.05 (Welch's $t$ test, FDR adjusted) are shown, along with the composition.

was no significant difference between subadults and adults. In addition, the proportions of infectious disease pathways (e.g., epithelial cell signaling in *Helicobacter pylori* infection) in the aged group were significantly higher than in the subadult and adult groups, but there was no significant difference between subadults and adult groups.

**Blood immune genes and relationship with gut microbiota.** We measured blood cytokine expression levels for CLEC4E, CD4, CD8$\beta$, IL-6, and TNF-$\alpha$ from 29 koalas and analyzed expression profiles with age. The expression of CLEC4E was significantly higher expression in the aged group (Fig. 7A). Instead, the Expression of CD8$\beta$ and CD4 was significantly higher in subadults (Fig. 7B and C). There were no statistically significant differences between groups in IL-6 and TNF-$\alpha$ expression. However, adult koalas exhibited a nonsignificant trend of higher expression of IL-6 and TNF-$\alpha$ (see Fig. S1) compared to subadult and aged koalas.

To reveal the interplay between the gut microbiota and the host immune system, we calculated the correlations between the gut microbiota (at the genus level) and the host immune markers. Overall, the three age groups have quite different correlations between gut microbiota genera and host immune factors. The gut microbiota of subadult koalas was closely correlated with CD8$\beta$ and IL-6, while little correlation with CLEC4E and CD4 (Fig. 7D). The gut microbiota of adult koalas was closely correlated with CLEC4E, CD4, and CD8$\beta$ but showed little correlation with IL-6 and TNF-$\alpha$ (Fig. 7E). The gut microbiota of aged koalas was closely correlated with CD4, IL-6, and TNF-$\alpha$, while little correlation with CLEC4E and CD8$\beta$ (Fig. 7F).

**Sex, reproductive state, and living environment.** Sex is the second factor affecting the gut microbiota of captive koalas, based on the percentage of taxa that are strongly related (8 to 30% of taxa at all taxonomic levels, Fig. 2). We found many bacterial species that differed in abundance by sex, second only to age (see Fig. S4A). Females had more *Verrucomicrobia* and *Planctomycetes* at the phylum level (particularly from class *Verrucomicrobiae* and *vadinHA49*). Females had more *CF231*, *Bacteroides*, *Akkermansia*, and *Tannerella* at the genus level. *Firmicutes* were found in greater numbers in males (particularly from class *Clostridia*). Furthermore, compared to males (median = 2.28), the B/F ratio was highest in females (median = 4.03; see also Fig. S4B). Sex had no significant effect on any of the alpha-diversity indicators and had little bearing on the beta diversity between samples (see Fig. S4C and Table 2). We discovered that sex was the second factor that impact the predicted function of the koala gut microbiota, just like our taxonomic study suggested. We found that the female group exhibited larger proportions in the pathways for glycan biosynthesis and metabolism, cofactor metabolism, and vitamin metabolism, whereas the male group exhibited higher proportions in the pathways for carbohydrate metabolism (see Fig. S5). Our results showed that male koalas exhibited a nonsignificant tendency toward increased expression of IL-6 and TNF-$\alpha$ (see Fig. S5C) compared to female koalas.

Only a small number of bacterial species were found to differ in abundance based on reproductive state (see Fig. S6A). Lactation females harbored more *Lactobacillus* than cycling females. Furthermore, the B/F ratio and beta diversity between samples were unaffected by the female reproductive state (Table 2; see also Fig. S6B). The gut microbiota of cycling females exhibited higher alpha diversity (Chao1) compared to lactation females (see Fig. S6C). There was no evidence that the predicted metabolic pathways varied according to reproductive status. Since we did not collect blood from lactating females, we do not know whether the reproductive status has any effect on the immune system.

The gut microbiota of koalas living in noisy (tourist regions, $n = 19$) and quiet habitats (not shown to tourists, $n = 10$) were also studied. Only a few taxa had varied abundances depending on their living conditions (see Fig. S7A). The living environment did not influence the B/F ratio between samples (see Fig. S7B). The living environment had no bearing on any alpha-diversity metric and beta diversity between samples (see Fig. S7C and Table 2). There was no evidence that the predicted metabolic pathways differed between noisy and quiet habitats. Interestingly, the expression of IL-6 was significantly greater in koalas living in noisy habitats than in koalas living in quiet habitats (see Fig. S7D).

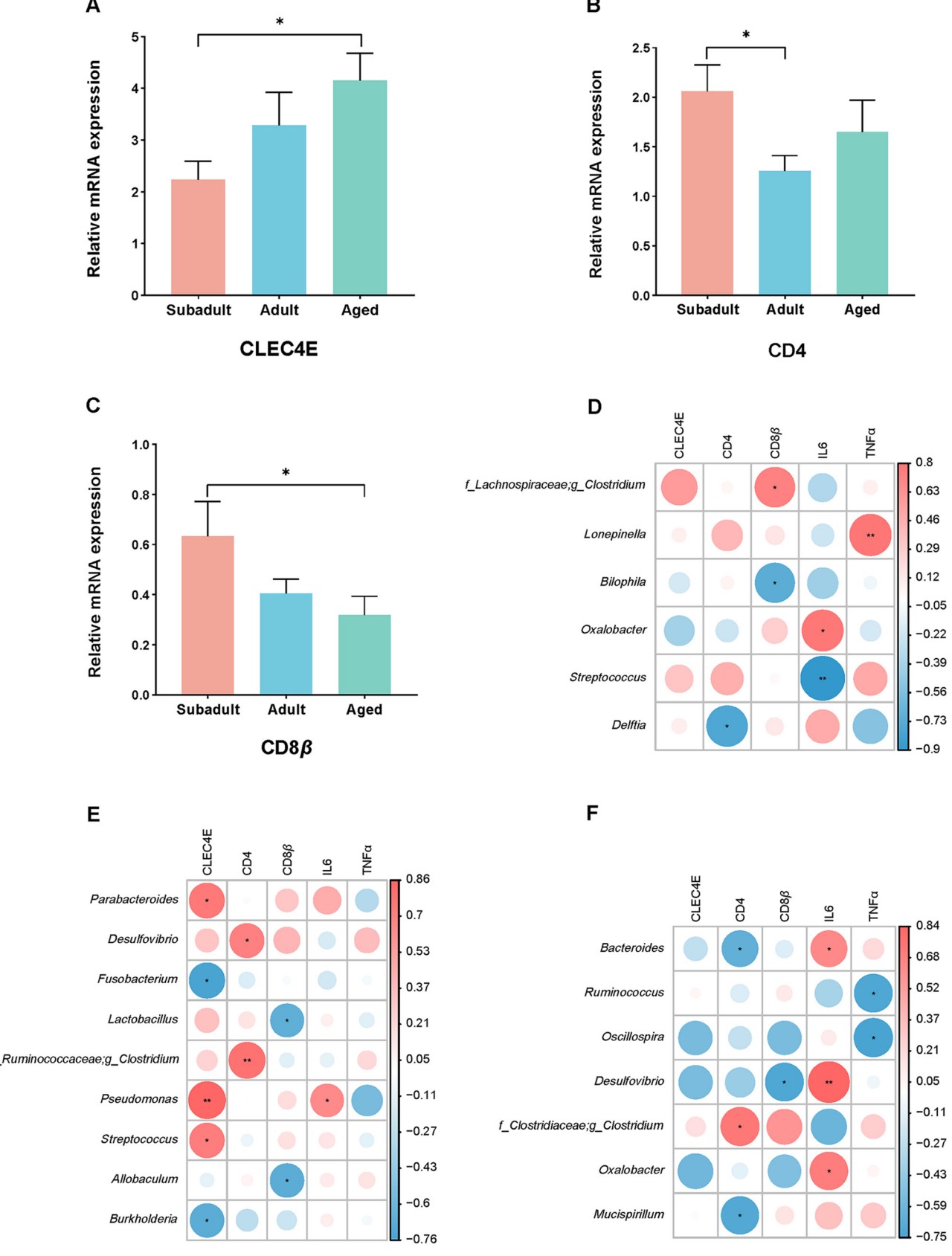

**FIG 7** Expression of blood immune gene influence by age. (A to C) Effect of age on the koala inflammatory cytokines: CLEC4E (A), CD4 (B), and CD8$\beta$ (C). Mean values ± the SEM are shown. The significance of the difference between groups was tested by ANOVA with an LSD *post hoc* test. Spearman

## DISCUSSION

We studied the role of host and environmental factors in influencing gut microbiota structure and the host immune system in a healthy captive koala population. Our findings implied that age affected koala microbiota composition and function the most, explaining a quarter of the variation of the koala microbiota, followed by sex, whereas other factors, such as reproductive status and environment, had little impact. We found that lipid metabolism, biosynthesis of other secondary metabolites, and infectious disease pathways were significantly correlated with age. Furthermore, we found that age impacts the host immune system and that different age groups have different correlations between gut microbiota genera and host immune factors, whereas other factors had no impact. These data show that disturbed habitat and captivity conditions do not appear to be stressors for captive koalas.

The koala gut microbiota has been shown to have many microbial species that have dynamic changes after birth and could play different roles (33). The age-associated microbiota discovered in captive koalas may play a role in the growth and health of the host. Depending on the direction of their age correlation, these microbiotas could play different roles. The makeup of the koala gut microbiota is essentially comparable to that of other mammalian gut microbiotas (33, 37), with bacteria from the phyla *Bacteroidetes* and *Firmicutes* being predominant (1, 38). *Akkermansia muciniphila* strains have been found in colostrum, breast milk, and young neonates and are generally regarded as helpful bacteria (39). Furthermore, *A. muciniphila* has been shown to improve the metabolic functioning and immunological responses of the host (40, 41). In contrast to earlier research (33), we discovered that the quantity of *A. muciniphila* varies between the three age groups. *A. muciniphila* abundance decreases with age and was highest in subadult koalas, whereas it declined in adult and aged animals. *Bacteroides fragilis* is a commensal Gram-negative obligate anaerobe that belongs to the *Bacteroides* family and maintains a complex and generally beneficial relationship with the host when retained in the gut, but when it escapes the gut it becomes an opportunistic pathogen (42–44). We discovered that aged koalas have more *B. fragilis* than adult and subadult koalas and the correlations between the gut microbiota and the host immune markers showed that *Bacteroides* in aged koalas showed a significant positive correlation with IL-6 (Fig. 7). Furthermore, *Lonepinella koalarum*, a Gram-negative bacterium found in koala feces, breaks down tannin protein complexes (45). This is a single species of the genus *Lonepinella*, which belongs to the *Pasteurellaceae* family. Here, the proportion of *L. koalarum* was highest in subadult koalas, with a negative correlation with age. We suggest that *L. koalarum* dominated the degraded tannin protein complexes in the subadult period, but that as the abundance of *L. koalarum* decreased in adults and aged periods, other methods of tannin degradation became dominant. However, we discovered that a bacterial species increased with age and was first reported in the feces of koalas in this study. *Ruminococcus flavefaciens*, a major plant cell wall-degrading bacteria, is the primary degrader of plant structural carbohydrates in the rumens of mammals (46–48). *R. flavefaciens*, we guess, dominates the bacterial breakdown of tannin complexes or fibers in adult and aged koalas.

The remarkable age-dependent changes, including the B/F ratio and beta diversity, emphasized the actual effects of age on the gut microbiota in captive koalas. The *Firmicutes/Bacteroidetes* (F/B) ratio is widely accepted to be involved in health-related conditions or diseases such as obesity (49, 50). In our study, *Bacteroidetes* was the most prevalent phylum, followed by *Firmicutes* and *Synergistetes* in each sample. So, we calculated the *Bacteroidetes/Firmicutes* (B/F) ratio. The B/F ratio was higher in subadult and adult koalas and decreased in aged koalas, resembling observations in humans or

**FIG 7** Legend (Continued)

correlations between gut microbiota abundances and the levels of host immune markers in blood samples from the three age groups are shown: subadult (D), adult (E), and aged (F). For each heat map, the rows correspond to gut microbiota taxa at the genus level, and the columns correspond to immune factors. The red and blue dots represent positive and negative correlations, respectively. The intensity of the colors denotes the degree of correlation between the genus abundances and the circulating levels of host blood immune factors. *, $P < 0.05$; **, $P < 0.01$.

other mammal animals (51, 52). As PCoA revealed, the gut microbiotas of samples within each group were remarkably similar, whereas different age groups clustered differentially. Captive individuals all had the same assigned diet, which explains why the compositions of gut microbiotas were quite similar at the same age (22, 53).

At the functional level, bacterial genes are involved in lipid metabolism and the biosynthesis of other secondary metabolites in adult koalas more than in subadult and aged koalas. Particularly, bile acid biosynthesis, fatty acid degradation, and sphingolipid metabolism became more prevalent, indicating that this was the time when both bacterial energy production and cellular activity were at their peak. The gut microbiota was strongly related to host metabolism (54). So, the increase in organismal metabolism reflects the richness of the gut microbiota system in adult koalas, which is consistent with the results of alpha diversity. However, under several conditions, an increase in bacteria that boost host metabolism may also stimulate inflammation or even inhibit immune response (55, 56). We showed here that adult koalas exhibited a trend toward higher expression of IL-6 and TNF-$\alpha$ or lower expression of CD4 and CD8$\beta$. Furthermore, bacterial genes are involved in infectious diseases in aged koalas more than in subadult and adult koalas, especially the epithelial cell signaling in *Helicobacter pylori* infection. On the one hand, *H. pylori* can influence the abundance and diversity of gut microbiota and be highly associated with cellular senescence (57, 58). On the other hand, *H. pylori* metabolites were reported that exacerbate gastritis through C-type lectin receptors such as C-type lectin domain family 4 member (CLEC4E) (59). This is consistent with the significantly greater CLEC4E expression observed in the aged group.

In addition, we investigated how the host immune system and the microbiota interact, as well as how age affects these interactions. More and more research has been conducted that reveals the immune system was a vital relationship with age and gut microbiota (31, 32, 60). Differences in the prevalence of infectious diseases across the age of the koalas were also noted (27, 61). CLEC4E is a surface marker expressed primarily on macrophages and acts as an activating receptor for cell wall components of bacterial species (28, 62). Previous studies have shown that significant upregulation of CLEC4E was seen in koalas with *Chlamydia pecorum* infection compared to those without infection (62). Our findings show that the expression of CLEC4E is highest in aged koalas, perhaps suggesting that aged koalas are at a higher risk of *C. pecorum* infection. Moreover, many microbial genera were highly related to the expression of functional lectins (63). Here, we found that the gut microbiota of adult koalas was closely correlated with CLEC4E. CD4 and CD8$\beta$ are signals to immune cells and also allow discrimination between classic helper T cells and cytotoxic T-cell families (28). According to research, CD4 and CD8$\beta$ levels were noticeably lower in koala retrovirus (KoRV)-positive koalas than in negative koalas (26, 64, 65). Compared to subadults, the expression of CD4 and CD8$\beta$ were decreased in adult and aged koalas in this study. We suggest that adult and aged koalas are at a higher risk of KoRV infection.

Individual variables such as sex, living environment, and female reproductive status showed only a minor impact on the gut microbiota and immune system (14). Although female koalas have more bacterial taxa than males, this is not the main factor that influences gut microbiota composition. At the functional level, females had higher proportions of metabolism pathways than males, especially glycan biosynthesis and metabolism and the metabolism of cofactors and vitamins. These pathways were mainly involved in many organismal functions, such as immune and inflammation regulation (66, 67).

There were several limitations in our research. Although the captive koalas' fecal microbiotas were similar to those found in wild koalas (1), our findings do not represent the entire species. Seasonal, spatial, lifestyle, and habitat environment factors all have an impact on animal gut microbiotas (14, 15, 36, 68).

In summary, we performed a comprehensive survey of the gut microbiota of different age koala populations. The gut microbiotas of koalas are typical for mammals and have a consistent core group of taxa. These core genera exhibited a remarkable change across

**TABLE 3** The gene sequence for RT-qPCR

| Target gene | Direction | Sequence (5′–3′) |
|---|---|---|
| CD4 | Forward | CCTGCCAAATTCTCCTTCCCTCTG |
| | Reverse | TCCACCTGCCACCTCAGTTCTC |
| CD8$\beta$ | Forward | AAGGTCACTCAACAGAATGGTTCCC |
| | Reverse | GATCAGCAAAATCACAGCACATCCC |
| CLEC4E | Forward | AGCAAAACCCAGTGGAAGAGAGTTC |
| | Reverse | TTGTTGAATGGCGTACCATCTACCC |
| IL-6 | Forward | GTGACGATAGCAATGAGGCACTAGC |
| | Reverse | ACAATCCTTGGCAAGCATGTCTCC |
| TNF-$\alpha$ | Forward | CTGCCTCTGCCTGTCACTTATCTTC |
| | Reverse | ATCTGTGGACTCCTCTTCCTTCTGG |
| GAPDH | Forward | GGCAAATTCAAGGGCACTGTCAAG |
| | Reverse | CAACATACTCGGCTCCAGCATCTC |

different age periods. Our results reliably show the individual (age, sex, and reproductive status) and environmental factors that may affect captive koala microbiota composition and function. In addition, we also discovered age-associated changes in inflammation or immunity and the correlations between gut microbiota and host immune markers. To effectively prevent or treat infectious illnesses, we must have a thorough understanding of the koala microbiota and how the immune system and microbiota interact.

## MATERIALS AND METHODS

**Study subjects and fecal sample collection.** We collected 35 fecal samples and 29 blood samples over 3 months between November 2021 and January 2022 from captive koalas (the largest koala population outside Australia) living in the Guangzhou Chimelong Safari Park in China (23°00′N, 113°33′E). A controlled environment with a temperature range of 15 to 26°C and a relative humidity range of 40 to 70% was used to keep all koalas. The koalas shared the same assigned diet. Various forms of data, such as identification, sex, body mass, reproductive status, and food information, were gathered for each sample taken from captive koalas. A total of 35 fecal samples were collected from different ages, sex, living environments, and reproductive states captive koalas. These groups included subadults (1 to 3 years old, 5 females and 4 males, $n = 9$), adults (4 to 7 years old, 5 females and 5 males, $n = 10$), older animals (9 to 13 years old, 5 females and 5 males, $n = 10$), and lactating females (3 to 7 years old, $n = 6$). Fecal samples were transported at 4°C and kept at $-80$°C in the lab until DNA extraction was performed. We collected 29 blood samples from different-aged animals, including subadults (1 to 3 years old, $n = 9$), adults (4 to 7 years old, $n = 10$), and aged animals (9 to 13 years old, $n = 10$). To prevent causing stress in lactating female koalas, we did not collect blood from them. Portions (500 to 1,000 $\mu$L) of blood were collected and stored at 4°C until RNA extraction and cDNA synthesis on the same day. This study was performed in accordance with the protocols of the Chimelong Safari Park and South China Agriculture University.

**16S rRNA sequencing and data processing.** After extracting microbial DNA, we used PCR primers to produce the V3 and V4 sections of the 16S rRNA gene (F, 5′-ACTCCTACGGGAGGCAGCA-3′ and R, 5′-GGACTACHVGGGTWTCTAAT-3′). QIIME2(2019.4) (69) was used for microbiota bioinformatics, with minor modifications made according to the official tutorials (https://docs.qiime2.org/2019.4/tutorials/). Using the demux plugin, raw sequence data were demultiplexed, and primers were cut using the cutadapt tool (70). Using the DADA2 plugin (71), the sequences were then quality filtered, denoised, and combined, and chimeras were eliminated. We analyzed microbial diversity in QIIME2, including taxonomic composition, alpha-diversity measures (Chao1 and observed species), and beta-diversity metrics (Bray-Curtis dissimilarity). The beta diversity values were compared using the nonparametric method PERMANOVA. Bray-Curtis distances were used to perform principal coordinate analysis (PCoA) of the bacterial communities. The linear discriminant analysis (LDA) effect size was used to identify differences in community composition in different sample groups (LEfSe).

**Isolation of total RNA and RT-qPCR.** A blood RNA isolate kit was used to extract total RNA from collected blood in accordance with the manufacturer's instructions (Biodai). HiScript III RT SuperMix for qPCR (+gDNA wiper; Vazyme) was used to create cDNA from total RNA (1 g), and a ChamQ universal SYBR qPCR Master Mix (Vazyme) was used to perform real-time quantitative PCR (RT-qPCR) amplification (Thermo Fisher Scientific, Inc.). Table 3 presents the primer sequences, and the $2^{-\Delta\Delta CT}$ method was used to analyze the data, which are presented as the gene expression to relative GAPDH.

**Statistical analysis.** First, we combined the counts at the taxonomic level (i.e., the number of reads per taxonomy and per sample). Only taxa whose average relative abundance over samples was $\geq 0.01$% were evaluated. Spearman's correlation analysis was used to find the correlations between ASVs and age as a continuous variable. Correlation analysis of gut microbiota in age and blood inflammatory

cytokines was performed using Spearman's correlation analysis. The differential abundance testing in age groups was verified by using a Kruskal-Wallis rank sum test, while sex, reproductive state, and living environment groups were verified by using the Mann-Whitney U test. The differential functional profiling of microbiota in each group was verified by using Welch's $t$ test with the false discovery rate (FDR) adjusted. The significance of the expression of blood immune genes between age groups was tested by analysis of variance (ANOVA) with a least-significant-difference (LSD) *post hoc* test, while sex, reproductive state, and living environment groups were tested by using a Student $t$ test. Differences were considered significant when the corrected $P$ value was <0.05. Statistical analysis was performed using IBM SPSS statistics 22.

**Data availability.** All 16S sequence data used in this study are available at the NCBI Sequence Read Archive (https://www.ncbi.nlm.nih.gov/) under BioProject number PRJNA871357. The data will be made public upon the acceptance of the manuscript.

## SUPPLEMENTAL MATERIAL

Supplemental material is available online only.
**SUPPLEMENTAL FILE 1**, PDF file, 1.4 MB.

## ACKNOWLEDGMENTS

We thank the Chimelong Group Co., along with the veterinary and animal care staff of the Chimelong Safari Park, for permission to conduct research and for ongoing support for our long-term research project.

Conceptualization, data curation, formal analysis, resources, investigation, and visualization (J.C.); methodology (J.C. and W.L.); collected samples (J.C., J.D., T.L., Z.W., and T.Z.), writing—original draft (J.C. and S.G.); writing—review and editing (all authors); funding acquisition (S.G. and G.D.); and supervision (X.Z., S.G., T.Z., P.Z., M.P., and G.D.). All authors read and approved the final manuscript.

We declare that we have no competing interests.

This project was totally supported by Guangdong Chimelong Philanthropic Foundation (5500-PH22008).

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
