## [Reviewer comments · Microbiology Spectrum]

Microbiology Spectrum

Age was a key effect on the gut microbiota and immune system in captive koalas (*Phascolarctos cinereus*)

Jiayan Chen, Weijie Lv, Xueli Zhang, Tianyou Zhang, Jian Dong, Zhihui Wang, Tingting Liu, Peng Zhang, Michael Pyne, Guixin Dong, and Shining Guo

Corresponding Author(s): Shining Guo, South China Agricultural University

Review Timeline:

Submission Date:	October 10, 2022
Editorial Decision:	November 9, 2022
Revision Received:	November 15, 2022
Accepted:	December 5, 2022

Editor: Yunhe Fu

Reviewer(s): Disclosure of reviewer identity is with reference to reviewer comments included in decision letter(s). The following individuals involved in review of your submission have agreed to reveal their identity: Yi Wu (Reviewer #1); Xiaolong Xu (Reviewer #2)

Transaction Report:

DOI: <https://doi.org/10.1128/spectrum.04101-22>

November 9, 2022

Prof. Shining Guo
South China Agricultural University
wushan road
guangzhou
China

Re: Spectrum04101-22 (Age was a key effect on the gut microbiota and immune system in captive koalas (*Phascolarctos cinereus*))

Dear Prof. Shining Guo:

Link Not Available

Sincerely,

Yunhe Fu

Journals Department
Reviewer comments:

Reviewer #1 (Comments for the Author):

In this manuscript, the authors describe the effect of individual and environmental in the gut microbiota and the host immune system of captive koalas. Demonstrated through many analyses and experiments that age was the key reason. I think the research is a carefully done study and the findings are of considerable interest. The logic and structure of the manuscript are clear. The results and conclusion are well illustrated and convincing. Even though the research question is very interesting, the study has some caveats:

Q1.The function of the bacteria should be described in the introduction or discussion instead of described in the abstract.

Q2.Where is the conclusion at the end of the abstract?

- Q3.Line 51, "several lignin ... in the gastrointestinal tract of koalas." Which bacteria have been identified?
- Q4.The introduction is not detailed enough. In the third paragraph, you talked about immune response and cytokine. The introduction of koala cytokine and its previous study is not explicit and needs to be supplemented.
- Q5.Where is the sex-related cytokines conclusion?
- Q6.Line 65, "Gut microbiota can also help the host detoxify secondary compounds from plants." Please give us some examples.
- Q7.Line 24, "gut microbial" should be "gut microbiota".
- Q8.Line 38, "gut flora" should be "gut microbiota".
- Q9.Line 83, "the gut microbiota ...an endangered animal." The language is not professional. Should be: the research in gut microbiota
- Q10.Line 106-111, table 3 "N" should be in italic.
- Q11.Line 264, "gut mycobiome" spelling mistake.
- Q12.Line 381, Define each abbreviation and introduce it in parentheses the first time it is used. e.g., "C-Type Lectin Domain Family 4 Member (CLEC4E)"
- Q13.The figures are not big and clear enough. All the figures need to be improved.
- Q14.Figure 1,2 needs to be marked A, B.
- Q15.Use the italic system with the scientific name of gut microbiota in the figure legend.
- Q16.Figure 5 demonstrated diversity analysis, So the legend title should be described as diversity differences.

Reviewer #2 (Comments for the Author):

This paper mainly describes the gut microbiota composition and the blood cytokine in different age captive koalas living in China. The author also analyzed whether sex, reproductive state, and living environment affected the captive koala. This study shows that individual factors are key factors in the growth and health of captive koalas. It is an interesting and original work. The reader will be interested in reading. However, there are some deficiencies that need to be corrected.

- Q1.The abstract is incomplete. I cannot find the main conclusion in the abstract.
- Q2.Does not completely describe the main points of your research in the importance part. The author need to addition describe the significance and the innovation points of the study.
- Q3.The authors described the importance of studying gut microbiota. But didn't describe the previous study about koala gut microbiota. Similarly, the authors just talk about the cytokines being suitable targets for assessing immune responses. Please supplement the previous study about koalas.
- Q4. The number of ASVs mean SD = 768.54{plus minus}132.94. And the sample total of 35 in this study. Please check if the total number of ASVs is correct.
- Q5.The format of the Latin name is incorrect. All the scientific names of gut microbiota should in italic. "β" in the text and "P" in figure legends, etc should in italic.
- Q6.Many English mistakes were found in this manuscript, eg. "gut flora", and "gut mycobiome" should be "gut microbiota".
- Q7.Line 298, How many live in noisy koalas or quiet habitats koalas respectively?
- Q8.Line 215, the initial letters need to be capitalized.
- Q9.The resolution of the figures should be improved.

Staff Comments:

Preparing Revision Guidelines

Please return the manuscript within 60 days; if you cannot complete the modification within this time period, please contact me. If you do not wish to modify the manuscript and prefer to submit it to another journal, please notify me of your decision immediately so that the manuscript may be formally withdrawn from consideration by Microbiology Spectrum.

This paper mainly describes the gut microbiota composition and the blood cytokine in different age captive koalas living in China. The author also analyzed whether sex, reproductive state, and living environment affected the captive koala. This study shows that individual factors are key factors in the growth and health of captive koalas. It is an interesting and original work. The reader will be interested in reading. However, there are some deficiencies that need to be corrected.

Q1. The abstract is incomplete. I cannot find the main conclusion in the abstract.

Q2. Does not completely describe the main points of your research in the importance part.

The author need to addition describe the significance and the innovation points of the study.

Q3. The authors described the importance of studying gut microbiota. But didn't describe the previous study about koala gut microbiota. Similarly, the authors just talk about the cytokines being suitable targets for assessing immune responses. Please supplement the previous study about koalas.

Q4. The number of ASVs mean SD = 768.54 ± 132.94 . And the sample total of 35 in this study. Please check if the total number of ASVs is correct.

Q5. The format of the Latin name is incorrect. All the scientific names of gut microbiota should in italic. “ β ” in the text and “P” in figure legends, etc should in italic.

Q6. Many English mistakes were found in this manuscript, eg. “gut flora”, and “gut mycobiome” should be “gut microbiota”.

Q7. Line 298, How many live in noisy koalas or quiet habitats koalas respectively?

Q8. Line 215, the initial letters need to be capitalized.

Q9. The resolution of the figures should be improved.

Dear Editor,

Thank you and reviewers for your valuable comments and suggestions on our manuscript, number: Spectrum04101-22, "Age was a key effect on the gut microbiota and immune system in captive koalas (*Phascolarctos cinereus*)". We have carefully checked and revised the manuscript accordingly. The Answer and Revisions are detailed as follows.

Responses to Reviewer's comments

Reviewers' comments:

Reviewer #1 (Comments for the Author):

In this manuscript, the authors describe the effect of individual and environmental in the gut microbiota and the host immune system of captive koalas. Demonstrated through many analyses and experiments that age was the key reason. I think the research is a carefully done study and the findings are of considerable interest. The logic and structure of the manuscript are clear. The results and conclusion are well illustrated and convincing. Even though the research question is very interesting, the study has some caveats:

Q1. The function of the bacteria should be described in the introduction or discussion instead of described in the abstract.

Answer and Revision: Thank you for your suggestion. The function of the bacteria has been described in the discussion. And we deleted the described function of the bacteria in the abstract.

Q2. Where is the conclusion at the end of the abstract?

Answer and Revision: Thank you for your suggestion. We have added the conclusion in the abstract (confer Lines 36-38 of the revised manuscript).

Q3. Line 51, "several lignin ... in the gastrointestinal tract of koalas." Which bacteria have been identified?

Answer and Revision: Thank you for your suggestion. We have added a description of

this (confer Lines 54-55 of the revised manuscript).

Q4. The introduction is not detailed enough. In the third paragraph, you talked about immune response and cytokine. The introduction of koala cytokine and its previous study is not explicit and needs to be supplemented.

Answer and Revision: Thank you for your suggestion. We have added a description of this (confer Lines 76-79 of the revised manuscript).

Q5. Where is the sex-related cytokines conclusion?

Answer and Revision: We have added a description of this (confer Lines 347-349 of the revised manuscript).

Q6. Line 65, "Gut microbiota can also help the host detoxify secondary compounds from plants." Please give us some examples.

Answer and Revision: Thank you for your suggestion. We have added a description of this (confer Lines 69-71 of the revised manuscript).

Q7. Line 24, "gut microbial" should be "gut microbiota".

Answer and Revision: Thank you for your suggestion. We have revised "gut microbial" to "gut microbiota".

Q8. Line 38, "gut flora" should be "gut microbiota".

Answer and Revision: Thank you for your suggestion. "gut flora" was revised to "gut microbiota".

Q9. Line 83, "the gut microbiota ...an endangered animal." The language is not professional. Should be: the research in gut microbiota

Answer and Revision: Thank you for your suggestion. We have revised this sentence as "the research in gut microbiota and blood cytokine levels provides more specific insights into the koala's unique biology without harming or disturbing an endangered

animal."

Q10. Line 106-111, table 3 "N" should be in italic.

Answer and Revision: Thank you for your suggestion. "N" in table 3 was revised in italic.

Q11. Line 264, "gut mycobiome" spelling mistake.

Answer and Revision: Thank you for your suggestion. "gut mycobiome" was revised to "gut microbiota"

Q12. Line 381, Define each abbreviation and introduce it in parentheses the first time it is used. e.g., "C-Type Lectin Domain Family 4 Member (CLEC4E)"

Answer and Revision: Thank you for your suggestion. We Define the abbreviation in Lines 435-436 and delete the introduction in Line 442 about CLEC4E.

Q13. The figures are not big and clear enough. All the figures need to be improved.

Answer and Revision: Thank you for your suggestion. All the figures were revised.

Q14. Figure 1,2 needs to be marked A, B.

Answer and Revision: Thank you for your suggestion. We have revised the Figure.

Q15. Use the italic system with the scientific name of gut microbiota in the figure legend.

Answer and Revision: Thank you for your suggestion. The scientific name of gut microbiota in all figure legends was revised in italic.

Q16. Figure 5 demonstrated diversity analysis, So the legend title should be described as diversity differences.

Answer and Revision: Thank you for your suggestion. We have revised the legend

title in Figure 5 as "Diversity differences in the gut microbiota among the three age groups."

Reviewer #2 (Comments for the Author):

This paper mainly describes the gut microbiota composition and the blood cytokine in different age captive koalas living in China. The author also analyzed whether sex, reproductive state, and living environment affected the captive koala. This study shows that individual factors are key factors in the growth and health of captive koalas. It is an interesting and original work. The reader will be interested in reading. However, there are some deficiencies that need to be corrected.

Q1. The abstract is incomplete. I cannot find the main conclusion in the abstract.

Answer and Revision: Thank you for your suggestion. We have added the conclusion to the abstract.

Q2. Does not completely describe the main points of your research in the importance part. The author need to addition describe the significance and the innovation points of the study.

Answer and Revision: Thank you for your suggestion. The significance and the innovative points of this study are to create the largest koala gut microbiota dataset in China to date and description several factors that may affect captive koalas on the gut microbiota and immune system, highlighting that age may be the key factor affecting captive koalas. Moreover, this study is the first to characterize the correlation between gut microbiota and cytokines in koalas. Better treatment strategies for infectious disorders may be possible if we can better understand the interactions between the immune system and the microbiome. We have added a description of this (confer Lines 40-46 of the revised manuscript).

Q3. The authors described the importance of studying gut microbiota. But didn't describe the previous study about koala gut microbiota. Similarly, the authors just talk about the cytokines being suitable targets for assessing immune responses. Please

supplement the previous study about koalas.

Answer and Revision: Thank you for your suggestion. The previous study about koala gut microbiota is described in the first paragraph of the introduction part. In addition, we follow the suggestion that supplemented the introduction of koala cytokine and its previous study (confer Lines 76-79 of the revised manuscript).

Q4. The number of ASVs mean SD = 768.54{plus minus}132.94. And the sample total of 35 in this study. Please check if the total number of ASVs is correct.

Answer and Revision: Thank you for your suggestion. We have carefully checked and made sure the total number of ASVs (after rarefaction) is correct.

Q5. The format of the Latin name is incorrect. All the scientific names of gut microbiota should in italic. "β" in the text and "P" in figure legends, etc should in italic.

Answer and Revision: Thank you for your suggestion. We have carefully checked and revised the whole manuscript and revised the "β" and "P" as italic.

Q6. Many English mistakes were found in this manuscript, eg. "gut flora", and "gut mycobiome" should be "gut microbiota".

Answer and Revision: Thank you for your suggestion. We have revised "gut flora" and "gut mycobiome" to "gut microbiota".

Q7. Line 298, How many live in noisy koalas or quiet habitats koalas respectively?

Answer and Revision: Thank you for your suggestion. The number of koalas living in noisy is 19 and in quiet habitats is 10). We have added a description of this in the text.

Q8. Line 215, the initial letters need to be capitalized.

Answer and Revision: Thank you for your suggestion. We have revised to capitalize the initial letters in this sentence.

Q9. The resolution of the figures should be improved.

Answer and Revision: Thank you for your suggestion. We have revised all the figures.

December 5, 2022

Prof. Shining Guo
South China Agricultural University
wushan road
guangzhou
China

Re: Spectrum04101-22R1 (Age was a key effect on the gut microbiota and immune system in captive koalas (*Phascolarctos cinereus*))

Dear Prof. Shining Guo:

Your manuscript has been accepted, and I am forwarding it to the ASM Journals Department for publication. You will be notified when your proofs are ready to be viewed.

Sincerely,

Yunhe Fu
Editor, Microbiology Spectrum
